# An Insight into the Prevention and Control Methods for Bacterial Wilt Disease in Tomato Plants

Sixuan Wu [1,2], Hao Su [1,2], Fuyun Gao [1,2], Huaiying Yao [3], Xuelian Fan [4], Xiaolei Zhao [5] and Yaying Li [1,2,*]

1 Key Laboratory of Urban Environment and Health, Institute of Urban Environment, Chinese Academy of Sciences, Xiamen 361021, China; sxwu@iue.ac.cn (S.W.)
2 Zhejiang Key Laboratory of Urban Environmental Processes and Pollution Control, Ningbo (Beilun) Zhongke Haixi Industry Technology Innovation Center, Ningbo 315830, China
3 School of Environmental Ecology and Biological Engineering, Wuhan Institute of Technology, Wuhan 430205, China
4 Ningbo Agricultural Technology Promotion Station, Ningbo 315800, China
5 Beilun District Agriculture and Rural Bureau, Ningbo 315800, China
* Correspondence: yyli@iue.ac.cn; Tel.: +86-0574-860-859-69

**Abstract:** Continuous cropping is the primary cultivation method in Chinese facility agriculture, and the challenge of it stands as a global issue in soil remediation. Growing tomatoes continuously on the same plot for an extended period can result in outbreaks of tomato bacterial wilt. It is caused by the soil-borne bacterium *Ralstonia solanacearum*, a widespread plant pathogen that inflicts considerable damage on economically significant crops worldwide. Simultaneously, this plant pathogen proves extremely resilient, as it can adhere to plant residues and persist through the winter, continuing to infect plants in subsequent years. Scientists have dedicated considerable efforts towards finding effective methods to manage this disease. This article delineates the characteristics of tomato bacterial wilt and the various types of pathogenic bacteria involved. It systematically reviews the progress in research aimed at controlling tomato bacterial wilt, encompassing both physical and biological aspects concerning soil and plants. Emphasis is placed on the principles and current applications of these control measures, alongside proposed improvements to address their limitations. It is anticipated that the future of tomato bacterial wilt control will revolve around the development of a novel environmental protection system and efficient control strategies, focusing on microecological management and enhancing tomato resistance against bacterial wilt through breeding.

**Keywords:** biological control agent; bacterial wilt; prevention and control; soil-borne diseases

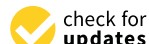

## 1. Introduction

Continuous cropping may result in reduced soil microbial diversity and an outbreak of soil-borne diseases [1], which have emerged as the primary factor limiting agricultural production and a significant contributor to crop health issues [2]. Tomato bacterial wilt is one of the main obstacles to tomato continuous cropping, and it is the second-largest plant bacterial disease in the world [3]. Understanding and managing bacterial wilt that devastates tomatoes, a major food staple crop worldwide, is of great importance. Tomato bacterial wilt has a wide impact on crops and many hosts, and more than 200 species of plants in more than 50 families have been infected by *Ralstonia solanacearum* [4].

In recent years, due to the rapid expansion of intensive agriculture, the risk of bacterial wilt has steadily increased, particularly under the practice of continuous single cropping and the influence of a warming climate [5,6]. Based on the symptoms, conditions, and rules of tomato bacterial wilt, this article summarizes the control measures and shortcomings of biophysical aspects of tomato bacterial wilt in recent years and puts forward corresponding improvement measures, aiming to come up with relevant improvement suggestions for the control of tomato bacterial wilt.

## 2. Symptoms of Bacterial Wilt and *Ralstonia solanacearum*

Bacterial wilt, induced by the soil-borne bacterium *R. solanacearum*, is a vascular disease of tomato plants [7]. A characteristic feature of the disease is that the leaves wither but remain green. As the disease progresses, black stripes appear on the plant's stem, the roots darken, the stem pulp decays, forming a cavity, and ultimately leading to the death of the plant [8]. *R. solanacearum* belongs to Gram-negative, aerobic, rod-shaped bacteria. It has the ability to persist in soil or plant disease residues for many years at an optimal growth temperature of around 32 °C. Under appropriate environmental conditions, *R. solanacearum* can infect the host through plant root wounds and cause diseases. *R. solanacearum* infects plant roots with soil as the medium and colonizes root wounds, root tips, secondary roots, etc. After a certain period of penetration, it reaches the vascular system of the plant, propagates in large numbers in the xylem of the plant, and secretes a large amount of extracellular polysaccharide, which obstructs the water transport inside the plant and eventually leads to the wilt and death of the plant [9]. The yield loss caused by bacterial wilt is generally 15–95%, which is an important limiting factor in the production of many crops and cash crops. As a landmark discovery, advances in the study of bacterial virulence factors have provided evidence that T3SS is one of the main pathogenicity determinants in *R. solanacearum* [7]. Based on variations in pathogenicity exhibited by *R. solanacearum* towards tomatoes, peanuts, tobacco, and eggplants, the pathogen can be categorized into three distinct pathogenicity types. Pathogenicity Type I strains display moderate pathogenicity towards all four hosts. Pathogenicity Type II strains exhibit strong pathogenicity towards tomatoes and eggplants, and moderate pathogenicity towards tobacco, but show no pathogenicity towards peanuts. Pathogenicity Type III strains demonstrate strong pathogenicity towards tomatoes and eggplants, with moderate pathogenicity towards peanuts and tobacco (Table 1).

**Table 1.** Pathogenicity of different *Ralstonia solanacearum* to plants.

| *Ralstonia* / Type of Plant | Tomato | Eggplant | Tobacco | Peanut |
|---|---|---|---|---|
| Pathogenicity Type I | Moderate | Moderate | Moderate | Moderate |
| Pathogenicity Type II | Strong | Strong | Moderate | No toxicity |
| Pathogenicity Type III | Strong | Strong | Moderate | Moderate |

At present, the control of bacterial wilt is a research hotspot in the crop cultivation field. Tomato bacterial wilt is difficult to control because of the diversity of pathogens and the wide range of hosts; *R. solanacearum* can survive and transfer in the deep soil for a long time, the chemical control effect is limited, and the long-term and irrational use of chemical pesticides will cause a series of environmental problems. In this article, the latest research progresses of tomato bacterial wilt control were reviewed from the aspects of breeding resistant varieties, biological control of bacterial wilt, and plant-immunity-induced resistance to provide a reference for control of the disease.

## 3. The Control of Bacterial Wilt by Soil Improvement

### 3.1. Plant Fungicide

There are abundant natural antibacterial substances in plants. These naturally active compounds are characterized by rapid decomposition and environmental friendliness. Studies have shown that the use of plant-derived antibacterial substances can inhibit the growth of *R. solanacearum* and alleviate or control the occurrence of disease. Natural compounds inhibit the proliferation of pathogenic bacteria in two ways (Figure 1): First, they destroy the cell wall and membrane structure of pathogenic bacteria, altering their bacterial morphology and affecting cell permeability. Second, they interfere with the expression and signal transduction of pathogenic proteins [10]. Tea polyphenols can significantly inhibit the proliferation of *R. solanacearum* and destroy the cell wall; it is speculated that the increased

permeability of the cell membrane leads to the exosmosis of metal ions and proteins, which further disrupts the cell metabolism and gradually destroys the cell structure [11]. Methyl gallate inhibits protein synthesis, succinate dehydrogenase (SDH) activity, and extracellular enzyme activity, rendering the bacteria non-virulent. At high concentrations, methyl gallate also hinders the respiration of *R. solanacearum* and disrupts the bacterial energy cycle metabolism [12]. Following treatment with 0.04% Palma Rosa, 0.07% Citronella, and 0.14% Eucalyptus essential oil, it was observed that, in comparison to the control group, all treatments except for the 0.04% Rose Grass treatment significantly reduced amino acids, nucleic acid purines and pyrimidine bases, carbohydrates, and lipids. After treatment with citronella, the degradation of cell components was the most pronounced, and changes in the cell structure, including the cell wall and cell debris, were observable [13]. Umbelliferone (UM, 7-hydroxycoumarin) is a phytoalexin found in the roots of sweet potatoes [14] and pharbitis nil [15]. It was found that UM suppresses the expression of T3SS regulators through the *PrhG–HrpB* and *HrpG–HrpB* pathways (*PrhG* and *HrpG* are the positive regulators of *HrpB*, which is essential for the pathogenicity of *R. solanacearum*) and inhibits the expression of many T3Es genes. Furthermore, UM suppressed biofilm formation without affecting swimming mobility, and bacterial populations of *R. solanacearum* were reduced by UM treatment in the roots and stems of tobacco. Additionally, UM reduced the virulence of *R. solanacearum* by suppressing biofilm formation as well as expression of T3SS regulators and T3Es genes, resulting in delayed tobacco bacterial wilt disease progression [16].

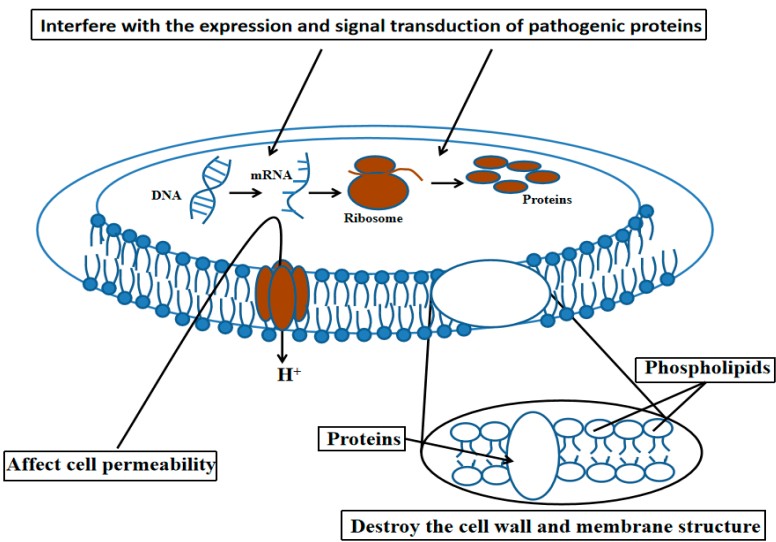

**Figure 1.** The mechanism of action of plant antibacterial compounds.

The extraction and processing technology of natural compounds is complicated, and the bacteriostatic mechanism is not clear. At present, there are relatively few plant-derived compounds applied to the control of *R. solanacearum*, and most of the reported studies are limited to pot experiments. At the same time, there are few plant-derived compound formulations for the control of bacterial wilt, and plant-derived agents are mainly insecticidal. Additionally, plant-derived agents are slow in effect and have poor efficacy against root bacterial diseases. In terms of improvement: The combination of chemical agents and plant source agents can enhance the immunity of plants to prevent bacterial wilt. In the form of a variety of agents, a comprehensive measure can be formed which can lead to fast sterilization. Due to the diversity of plant antibacterial components, new and environmentally friendly pesticides can be developed by using plant-derived active compounds.

### 3.2. Agricultural Antibiotic

Agricultural antibiotics are secondary metabolites produced by microbial metabolism that can inhibit or kill pathogenic bacteria. They are widely used, and there are 24 kinds

of agricultural drug resistance registered in China, with an annual output of more than 80,000 tons and an output value of 2.3 billion barrels. The main types of antibiotics are zhongshengmycin, streptomycin, hydromycin, etc., and it is not easy to produce environmental resistance and detriment. It can not only control plant diseases by acting on the cell structure of the pathogen, affecting its material and energy metabolism, but also improve the disease resistance genes of the plant [17]. Zhongshengmycin is an agricultural antibiotic derived from a Hainan strain of *Streptomyces* violaceus, which has been used in China to prevent tomato bacterial wilt [18], and it achieves a bacteriostatic effect by inhibiting peptide bond formation during the protein synthesis process of *R. solanacearum* [19]. Additionally, these antibiotics can stimulate the production of phytoalexin and lignin precursors in plants, thereby enhancing their disease resistance. When cultured for 7 days, using a mixture ratio of streptomycin sulfate and antagonistic fermentation solution at 3:2, the inhibition rate of *R. solanacearum* reached 84.7%, demonstrating a significant inhibitory effect [20]. In agricultural production, antibiotics such as zhongshengmycin, normycin, agrostreptomycin, and oxytetracycline hydrochloride have demonstrated positive outcomes in the control of tomato bacterial wilt. Normycin is a product of the fermentation of bacteria, and its mechanism is to interfere with the activity of metabolic bacteria and block the synthesis of bacteria protein to play a bactericidal effect. It has high efficiency and a broad spectrum and has a synergistic effect on disease prevention and control [21]. Coronatine (COR) is a compound produced by *Pseudomonas* syringae and has a structure similar to Jasmonic Acid-Isoleucine (JA-Ile). There are studies that have revealed that COR regulates tomato resistance to *R. solanacearum* by affecting the expression of key genes in plant hormone signal transduction pathways in plant–pathogen interaction pathways, such as inducing upregulation of JA synthesis genes and inhibiting gene expression in plant photosynthesis.

Most enterprises are short of research and development capabilities, so agricultural antibiotics lack effective marketing strategies. However, most research institutes focus on biological pesticide-related basic research and pay insufficient attention to the problems in the process of industrialization [22]. Chemical pesticides still occupy the vast majority of the market share, biological pesticides do not have the advantages of broad spectrum and quick effect, and there is also no price advantage. The operation and application requirements of agricultural antibiotics are too complicated, so farmers tend not to use them [17]. In addition, overuse of biological pesticides is vulnerable to environmental factors. In terms of improvement, with the deepening of the research on biocontrol factors, the future development of biological pesticides should be diversified. The application range of biological pesticides needs to be expanded, so the types and dosage forms should be suitable for different diseases and different crops [23]. Beyond that, we can carry out publicity campaigns from the long-term development of agricultural planting, so that farmers realize that microbial pesticides can achieve the long-term investment effect and eliminate farmers' concerns about the cost of microbial pesticides.

### 3.3. Avirulent Rasltonia solanacearum

The virulence differentiation of *R. solanacearum* is severe [24–26], and it can differentiate into virulent strains and avirulent strains. The former infect plants and cause plant disease, while the latter invade plants but do not cause disease [27]. In general, the use of closely related bacteria that are similar to *R. solanacearum* relies on mutual inhibition of bacteriocin and competition between the two bacteria for nutrient sites, resulting in inhibition of *R. solanacearum* [28]. Avirulent *R. solanacearum* has been employed to develop plant vaccines, which are inoculated in advance during the seedling stage. The strains will proliferate within the plant and induce the plant to produce disease-resistance responses. These include increased defense enzyme activity, the production of disease-resistant compounds, and enhanced expression of defense genes [29]. *Hrp* genes, required to set up a functional T3SS, are necessary for disease development in susceptible plants and elicitation of the hypersensitive response in resistant plants. The nonpathogenic Δ*hrpB* mutant

of *R. solanacearum* could induce the expression of resistance-related genes in *Arabidopsis thaliana*, with 26% of them upregulated, particularly those associated with abscisic acid biosynthesis and signaling [29]. However, simultaneous root inoculation by both the wild-type and *hrp* mutant strains did not induce protection, although the mutant strain was favored by a high mutant-to-wild-type strain inoculum ratio. These results suggest that protection may not be solely due to spatial competition between the two strains, as previously proposed [4]. The delay required between *hrp* mutant and wild-type strain inoculations suggested that some plant signaling pathways had to be established before inoculation of virulent bacteria. Heat-killed *hrp* mutant bacteria were also able to induce resistance but to a lower extent than live ones, which suggested that an active metabolism for both partners was required for full protection [30]. Some studies have shown that avirulent *R. solanacearum* has no direct inhibitory effect on *R. solanacearum* but could colonize plants and prevent the proliferation of *R. solanacearum* after pre-inoculation, thus delaying the occurrence of bacterial wilt and ultimately controlling the damage of bacterial wilt.

However, some research has claimed that the mutant could only achieve a certain effect on the prevention of tomato bacterial wilt, temporarily delaying the onset of tomato bacterial wilt, but it could not be eradicated [28]. The avirulent *R. solanacearum* colonized plants only briefly (28 days), which might be due to the existence of endophytic microorganisms that inhibited the proliferation of the bacteria, and it might also be related to nutrition and environment [31]. With the extension of time, the virulent strains have a stronger growth ability than avirulent strains. As for the solution, the avirulent *R. solanacearum* fermentation solution needs to be supplemented to increase the number of avirulent *R. solanacearum* in the plant. The other solution is to apply the agricultural streptomycin once or twice in two weeks after the initial application of this avirulent *R. solanacearum* strain, which helps to achieve the effect of continuous disease control. It can enhance the colonizing ability of non-pathogenic *R. solanacearum* in plants and make the control effect more lasting.

### 3.4. Microbial Inoculant

The application of microbial inoculants can induce a healthy rhizosphere microflora by altering the soil microflora, thus inhibiting tomato disease [32]. At present, most microorganisms related to plant disease inhibition are *Streptomyces*, *Pseudomonas*, and *Bacillus*. Antagonistic bacteria can inhibit or kill pathogenic bacteria, either directly or indirectly, through mechanisms such as competition for nutrients and spatial sites, induction of plant resistance, and secretion of antimicrobial metabolites (Figure 2). The proportion of bacteria and fungi is an important indicator of soil microbial ecology [33]. Soil microbial community structure and function is the core of the resistance of soil; soil microbial imbalances are considered to be one of the major causes of continuous cropping [34]. The action mechanism of biocontrol bacteria can be divided into two categories, namely direct inhibition mechanism and indirect inhibition mechanism. Direct inhibition mechanisms include antagonism, hyperparasitism, competition, and bacteriolytic action, while indirect inhibition mechanisms include promoting plant growth and inducing plant resistance. Many studies have shown that these mechanisms often do not operate in isolation, but it is likely that two or more mechanisms coordinate and act together [35]. *Pseudomonas putidosus* Pp17 isolated from potato rhizosphere soil showed strong antagonistic activity against potato *R. solanacearum*, and the control effect of the conservatory pot was up to 51.50% [36]. *Pseudomonas* can be used as a biocontrol microorganism and mainly has the following advantages: fast growth rate, suitable for large-scale fermentation production; quick use of seed and root exudates; strong colonization and reproductive capacity, and it can synthesize beneficial secondary metabolites, such as antibiotics, iron carriers, and other promoting biomass; and strong ability to adapt to environmental pressure [37]. *Bacillus subtilis*, *Bacillus amyloliquefaciens*, and *Bacillus licheniformis* could inhibit the occurrence of bacterial wilt, and *Bacillus subtilis* had the strongest inhibitory capacity, which could reduce the infection rate by 80%. *Streptomyces* is a type of actinomycete used in the industrial production of antibiotics, and more

than 70% of antibiotics in nature were produced by streptomycin bacteria, which could effectively prevent tobacco bacterial wilt [38]. In addition to a single antagonist strain, two or more compound antagonists can also be used, and the effect is better than that of a single antagonist [39]. It was found that the application of compound antagonists could increase the α and β diversity of tomato rhizosphere bacteria, improve the relative abundance of potentially beneficial bacteria groups (*Sphingomonas*, *Pseudomonas*, etc.), and reduce the relative abundance of *R. solanacearum* [40].

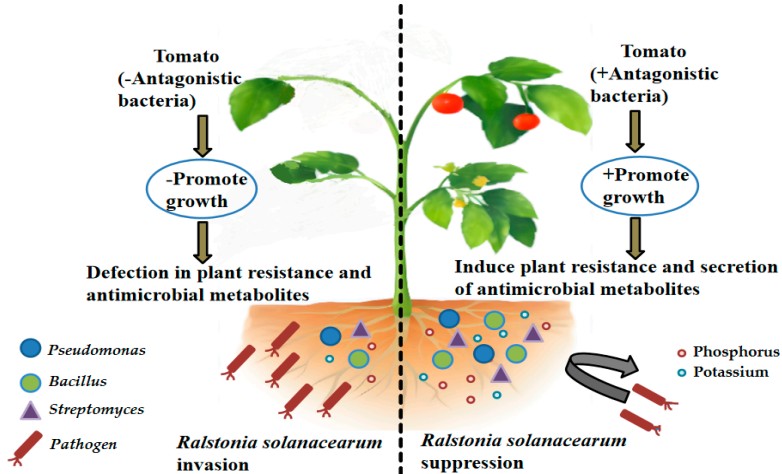

**Figure 2.** The mechanism of antagonistic bacteria.

Microbial inoculant has the advantages of being economical, environmentally friendly, safe, and sustainable, but also has the limitations of slow effect, strong specificity, instability, and its use is often greatly affected by the environment. The mechanism of bacterial wilt control and whether the microbial agent can be guaranteed after application in the soil need to be further studied. Additionally, it is important to explore the potential synergistic effects between different functional microorganisms, and the possibility of using more than two antagonistic microorganisms should be considered [38].

*3.5. Phages*

Since the discovery of phages, phage therapy has been developed, and its use as an environmentally friendly plant vaccine has made important advances, being widely used to treat human and animal diseases, as well as to suppress plant bacterial pathogens. Phages are greatly different between strains and have no complete cell structure. It is necessary to adopt different preservation methods [41], and ecological therapy has attracted more and more attention because of its advantages such as strong targeting, fast cracking speed, and small disturbance to the environment. Phages are a general term for bacterial viruses in which virulent phages can target host-specific receptors and lyse the host bacteria. According to this characteristic, virulent phages capable of obligate infection of pathogens can be selected as biocontrol agents, which can reduce the number of pathogenic bacteria and reduce the incidence of disease [42]. The colonization and survival of phages in the soil are the basis of reducing soil biological pollution. To suppress the proliferation of soil pathogens, sufficient concentrations of phages must be ensured. The higher the number of phages around the target pathogen, the contact and interaction between them will increase, and the greater the success rate of infection [43,44]. The factors that affect phage therapy can be divided into three categories: host bacterial profile and population size of phages, polymorphisms of pathogenic bacteria, and environmental factors that affect phage–pathogenic bacteria interaction, including soil temperature, pH, structure, nutrients, multiple pollutants, etc. When the *R. solanacearum* M4S strain was administered together with a phage, the incidence decreased from 95.8% to 14.5% in the control group [45,46]. The filamentous phage φRSM3 could control bacterial wilt caused by *R. solanacearum*.

The cell infection containing φRSM3 could enhance the expression of proteins related to the disease course of host plants, and the pretreated tomato could be protected from the infection by virulent pathogens [47]. The phage itself is harmless to the environment and the human body, so it is an important research direction of bacterial wilt control. The phage isolated from river water could control plant bacterial wilt by irrigation water under natural conditions [48]. Meanwhile, it was found that there was a significant negative correlation between the *R. solanacearum* growth and resistance under rhizosphere phage stress, indicating that the stronger the *R. solanacearum* resistance, the weaker the pathogenic ability of the *R. solanacearum* [49].

Phages are environmentally safe, but at the same time, their therapeutic spectrum is narrow, and they can only infect one or a few strains. Since bacteria have a variety of mechanisms to resist phage infection [50], the effectiveness of using a single phage in practical applications may be unstable. At the same time, the timing and dosage of phage therapy are difficult to control. In terms of improvement, to further explore different plants and farming methods, to overcome the resistance of pathogenic bacteria to phages and prolong the therapeutic effect of phages, two or more kinds of phages need to be used at the same time in clinical and agricultural disease prevention, namely phage cocktails [51].

### 3.6. Anaerobic Disinfection

Anaerobic soil disinfestation (ASD) is a management strategy that utilizes organic amendments to control soil-borne plant pathogens. It is an ecological environmentally friendly substitute for chemical fumigation, commonly referred to as a biologic soil disinfestation. The process of ASD involves the incorporation of organic amendments into the soil, followed by the saturation of the soil with water and the subsequent application of a plastic film to create anaerobic conditions (Figure 3). This treatment is typically maintained for a period of 3 to 5 weeks. ASD has been found to effectively eliminate several types of organisms present in soil, including bacteria, oomycetes, fungus, nematodes, and weeds [52]. The mechanism underlying soil anaerobic disinfection for the control of soil-borne diseases remains elusive. The primary proposed mechanisms that have been suggested are as follows: (1) generating metal ions, lowering soil pH, oxidation–reduction, and redox potential, rendering the soil environment unfavorable for the survival of soil-borne pathogens, thereby reducing the occurrence of soil diseases [53,54]; (2) the production of sterilization by-products, such as volatile organic acids or fermentation acids and the gas production, which can alter the composition of bactericidal and fungal communities in the soil [55,56]; and (3) the modification of the microbial species diversity and abundance; for example, ASD can induce changes in soil bacteria microbial functions such as denitrification, nitrogen fixation, and the production of organic acids through fermentation [57]. It was found that the application of a large number of organic materials, together with flooding, proved to be an effective method for quickly restoring the soil in a greenhouse vegetable field that had experienced significant degradation as a result of acidification and secondary salinization. At the same time, the challenges associated with continuous cropping, such as soil-borne pathogens and degradation of physical and chemical properties, can be mitigated. However, it is evident that the effects of different organic materials are obviously different [58]. Adding rice bran, wheat bran, and peanut bran in the treatment of ASD has been found to exhibit a preventive and controlling effect on bacterial wilt, with an efficacy above 90%. Furthermore, the addition of these bran types greatly reduced the population of *R. solanacearum* in soil by 31.3–97% [59]. ASD treatment groups exhibit a notable ability to inhibit the occurrence of bacterial wilt, potentially due to its direct inhibitory effect on the population of pathogenic bacteria. It is reported that ASD treatment can significantly reduce the population of bacterial *R. solanacearum* in soil up to 99.4% [60].

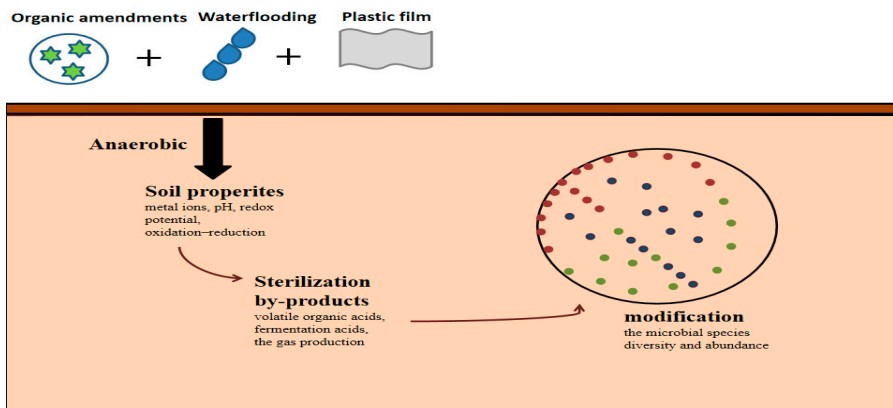

**Figure 3.** The mechanism of anaerobic disinfection.

The soil disinfection industry has emerged as an important part ensuring the sustainable development of agricultural production due to its high efficiency and economic attributes. ASD has been found to produce a volatile mixture that significantly inhibits the proliferation of *R. solanacearum*. However, the specific composition of this mixture remains unidentified. In addition, some fatty acids and low-priced metal ions were also produced during the process. It is imperative to do further research on the inhibitory effects of these substances on *R. solanacearum*. It is advisable for farmers to prioritize the utilization of straw as a carbon source when employing ASD technology. It is also recommended to refrain from using a mixed carbon source consisting of chicken manure and straw in conjunction with ASD technology. This approach is crucial in ensuring the long-term viability and sustainability of agricultural production [61].

## 4. Control of Tomato Bacterial Wilt by Improving Plant Traits

### 4.1. Selection of New Resistant Varieties

Long-term planting of a single tomato variety can result in an increased susceptibility to diseases and decline in overall quality. Different tomato varieties exhibit varying levels of resistance to bacterial wilt pathogens. The implementation of breeding and planting resistant varieties can effectively reduce the degree and rate of disease occurrence. Diseases caused by soil-borne bacteria, such as bacterial wilt and soft rot, pose significant challenges in terms of control. Consequently, breeding disease-resistant varieties is a viable strategy to mitigate the impact of these diseases. The traditional method for identifying resistance to bacterial wilt generally involves conducting experiments in soil that is infected with the pathogen. This allows for the observation of resistance in plants during their whole growth period. Subsequently, the process of inoculation progressed from the pot seedling stage to hydroponic inoculation. At present, hydroponic inoculation at the seedling stage is the most widely used method for identifying resistance [62]. In recent years, with the increasing demand for high-quality resistant tomatoes against bacterial wilt, a large number of resistant plants have been screened. Hideyoshi et al. conducted a screening of tomato LNSR-7 using cell culture to identify high resistance to bacterial wilt [63]. Transgenic tobacco plants overexpressing the *GmERF3* gene showed increased resistance to the bacterial pathogen *R. solanacearum*, which exhibited significantly reduced disease lesions compared with wild-type controls [64]. *GmERF3* belongs to a novel ERF class IV, which is typified by a conserved N-terminal signature sequence (MCGGAII/L). Previous research had demonstrated that overexpression of *ERF* genes enhances resistance to biotic and abiotic stresses. The *NtVQ35* gene of tobacco has a negative effect in resistance to bacterial wilt. There was a study showing that the tobacco *NtVQ35* gene-edited plant obtained using CRISPR technology was not prone to develop bacterial wilt compared to the wild type [65].

We should note that resistant varieties do not have permanent disease resistance. The resistance of tomato varieties against bacterial wilt is subject to instability in actual

production, mostly due to the interaction mechanism of plant pathogens, differentiation and variation in pathogenic strains, environmental changes, and other influencing factors. Under very favorable environmental conditions for disease occurrence, the resistant varieties possessing genes encoding resistance cannot be completely protected. The breeding of bacterial wilt resistant varieties is affected by many factors. For example, the source of resistance, the genetic relationship between resistance and other agronomic traits, the variability and differentiation of pathogens, plant and pathogen interaction mechanism, and breeding methods, which have been discussed in previous studies [66]. Therefore, the market acceptance of bacterial-wilt-resistant varieties can be influenced by spatial and temporal parameters, which in turn impact both yield and quality. Using gene-editing technology, plant genomes can be modified without introducing foreign DNA fragments. This has made it easier for the market to accept gene-edited crops and for scientists to avoid genetic transformation in many cases. In order to achieve more effective control measures, farmers must carefully choose disease-resistant tomato varieties, taking into consideration the specific environmental conditions and types of pathogens prevalent in different places.

### 4.2. Improvement of Tomato Plants by Hybridization

It is a universal phenomenon in the biological world that hybridization produces superiority. Therefore, using disease-resistant plants as parents, several hybrid combinations were designed with the screening methods of artificially adding pathogenic bacteria to produce high-quality and high-yield hybrid plant varieties, which also have resistance to bacterial wilt. The analysis of the genetic pattern of the bacterial-wilt-resistance phenotype is the basis of disease-resistance breeding. Only by clarifying the genetic inheritance law of bacterial wilt resistance can we avoid the blindness of resistance breeding, improve the efficiency of breeding resistant varieties/hybrids, and lay a theoretical foundation for disease-resistance breeding. Researchers have also used traditional breeding methods to develop disease-resistant new germplasm. The Yifeng No. 2 tomato, as reported by Qiu et al. [67], was developed by hybridization and exhibited the characteristics of resistance to bacterial wilt and strong adaptability. According to the identification of disease-resistance inoculation during the artificial seedling period, the average incidence rate was 10% after 5 weeks' inoculation of *R. solanacearum*, showing high resistance to bacterial wilt. The Gangyu NO. 1 tomato was a hybrid of GS619-54 and ZX1-53, and exhibited a bacterial wilt disease index of 16.8, showing its resistance to bacterial wilt [68]. The Yuekeda 301 tomato, a new combination of cherry tomato, was inoculated with *R. solanacearum*. The experimental findings revealed a bacterial wilt incidence rate of 40.37%, indicating a moderate level of resistance [69]. The new varieties of tomato, namely Jindi 363, were developed through hybridization of FP09-125 and FP055. The regional experiments were carried out in multiple regions, resulting in enhancements in both disease resistance and yield [70].

Although hybridization has many advantages, it also has some disadvantages. For example, hybridization will not produce new genes; all of its traits come from existing genes. The hybrid offspring also have character separation, and the breeding process is slow and complicated. The advantages of hybrid plants usually last only one generation, and they are not passed down to future generations. Farmers need to buy new seeds every year, which has a cost burden. Furthermore, hybrid plants have a higher demand for fertilizer. To realize their high-yield potential, farmers need to provide an adequate supply of fertilizer, which also increases the cost of growing hybrid tomatoes. With a comprehensive understanding of the advantages and disadvantages of hybrid tomatoes, we can make better use of them and bring more economic and environmental benefits to farmers.

### 4.3. Plant Grafting

Grafting is a distinct cultivation technique which differs from traditional tomato cultivation. It is a method of propagating plants asexually that involves pieces of different plants affixed together, resulting in their eventual fusion and subsequent growth as a unified

plant entity. Grafting technology has been shown to effectively cure soil-borne diseases and improve yield in tomato production. It requires grafting the upper portion of the desired plant trait onto a disease-resistant rootstock. The incidence of tomato bacterial wilt varies according to the specific rootstocks used in the grafting process. In recent years, many researchers have devoted themselves to screening rootstocks exhibiting high resistance to bacterial wilt. Grafting technology has the characteristics of high efficiency, low cost, and short time [71]. Grafted plants can absorb more nutrients and have increased tolerance to abiotic stresses such as extreme temperature, salinity, alkalinity, drought, flooding, and heavy metals [72]. Consequently, it can effectively improve plant resistance to disease and stress [73]. Studies had shown that the disease resistance of grafted tomato plants was closely related to the enhancement of tomato defensive enzyme activities, such as peroxidase isoenzyme, polyphenol oxidase isoenzyme, and catechol oxidase isoenzyme. Grafting has been a well-established and efficient technique in regions where bacterial wilt is prevalent in the tomato crop [74]. The rootstock 'Hawaii 7996' was grafted with several other tomato varieties with good affinity, and the results showed that the grafted seedlings had good resistance to bacterial wilt [75]. The tomato rootstock 'Guizhun No. 1' was subjected to grafting with the 'Da Mingxing' tomato, and it was found that the grafted plant had an abacterial wilt resistance rate above 90%. The incidence of bacterial wilt was found to be lower than 15% in both the grafted plants with scion and rootstock grafted plants [76]. Another study reported that resistant eggplant rootstocks reduced the death rates of grafted plants to 0–2.8% compared to the considerably higher mortality rate of 92.8% in non-grafted plants [77].

The compatibility of the scion and rootstock determines the success or failure of grafting. Notaguchi et al. successfully demonstrated the feasibility of cross-species grafting by using tobacco as an intermediary, *Arabidopsis* as the rootstock, and tomato as the scion [78]. In addition to affinity, the success of grafting also depends on some factors, such as conducive growth environment for the grafted plant and the ability of genetic material to be normally transported between scion rootstocks, which subsequently determines whether the grafted seedlings can achieve the goal of resistance to disease [79]. The application of grafted tomato plants is currently limited in scope, and grafting can change the texture of the resulting tomato fruits. It is necessary to properly promote tomato grafting technology in order to disseminate this technology among farmers. By doing so, the agricultural community may effectively mitigate the incidence of disease, increase crop production, and reduce planting costs, consequently encouraging the use of grafting technology. Given these circumstances, it is important to strengthen the excavation and breeding of wild tomato resources, while simultaneously cultivating more excellent rootstock varieties [80].

*4.4. Immune Resistance Inducer*

The most optimal, cost-efficient, and long-lasting method to sustain tomato productivity is through the cultivation of tomato varieties that possess genetic resistance to bacterial wilt [81]. However, it is worth noting that bacterial-wilt-resistant tomato varieties can be infected by emerging pathogenic strains, as well as the influence of climate change factors such as high temperature and humidity, excessive soil moisture, root damage, and $CO_2$ concentration in the environment. Consequently, these environmental conditions can lead to the loss of disease resistance in previously resistant tomato varieties [82]. In addition, induced resistance has proven to be a highly effective way to prevent and control plant diseases, especially in cases where disease-resistant varieties and effective chemical agents are not readily available in agricultural production. Induced resistance is an effective approach to enhance plant disease resistance by enhancing plant stress resistance and disease-resistance potential, hence resulting in long-lasting and comprehensive resistance against many pathogens [83]. Resistance can be induced in several ways. In the context of plant immunity, the induction and stimulation of immunological resistance mostly occur through external inducers or elicitors. Elicitors, which are a special class of compounds, have the ability to activate the host plant to produce a defense response. The avirulent *R.*

*solanacearum* strain, which was obtained by ultraviolet-light mutagenesis, exhibited notable efficacy in inducing resistance against tomato bacterial wilt. Recent research has revealed a significant interaction between plant immune system and rhizosphere microorganisms. These rhizosphere microorganisms can affect the expression patterns of immune-system-related genes in host plants, hence impacting plant metabolism to a certain extent [84]. It can also be induced by rhizosphere growth-promoting bacteria. The rhizosphere growth-promoting bacteria can stimulate and activate the plant defense response, thus inducing resistance. Furthermore, these bacteria can facilitate the transmission of this defensive reaction to other parts of the plant, thereby enhancing the plant's ability to resist the pathogen infections. Studies have found that the use of coronin has the potential to reduce the symptoms associated with tomato infection caused by *R. solanacearum*, improving the resistance of plants. A new immune resistance inducer derived from *Paecilomyces variotii* has been identified, exhibiting remarkable efficiency and environmental protection. This inducer possesses the ability to enhance plant growth, improving resistance to disease and increasing crop yield [85]. Benzothiadiazole (BTH) and its metabolites in plants have no bactericidal activity; rather, they can affect multiple aspects of the pathogen's life. BTH not only plays a role in signal transduction by simulating salicylic acid but also induces plant resistance by changing the activity of peroxidase (POD), phenylalanine aminolysase (PAL), and β-1, 3-glucanase (GLU), as well as influencing the expression of pathogenesis-related genes to produce PRP (disease-course-related protein) [86,87].

Plant immune inducers activate the plant immune system instead of directly acting on pathogenic microorganisms. Although immune resistance inducers have many advantages, they also need to be improved. First of all, it is still necessary to further investigate more effective novel immune resistance inducers. At present, immune resistance inducers still have the shortcomings of hysteresis and high cost. Secondly, the precise activation mechanism and mode of action of some immune resistance inducers in the context of crop disease resistance remain unclear [88]. It is very important to clarify the molecular basis governing the development of plant immune resistance inducers and explain how plant immune resistance inducers specifically improve the resistance of crops against disease. Additionally, it involves identifying the target of plant immune resistance inducers, receptor recognition, and key activation sites, subsequently employing new research methodologies and processes to develop highly applicable reagents.

## 5. Summary and Prospect

The utilization of physical and biological methods for managing soil and plant health is considered to be the economical and effective way to control tomato bacterial wilt. So far, in spite of the significant prevention and treatment of bacterial wilt, there exist certain limitations in both physical and biological control methods employed from soil and plant sources. There are some challenges in biological control, such as high control cost, complicated operation and unstable control efficiency. The ongoing evolution of pathogenic bacteria, resulting in the emergence of novel physiological variants, poses a significant challenge to the development of new disease-resistant types. This rapid evolution undermines the effectiveness and sustainability of breeding efforts aimed at enhancing disease resistance. Future research should also prioritize the examination of abiotic factors that contribute to reduction of bacterial populations. In the future, the biological and physical control methods for the management of tomato bacterial wilt can be integrated to obtain more effective measures. Most soil control measures for bacterial wilt not only target pathogenic bacteria but also have the ability to increase community diversity, readjust the structure of rhizosphere bacterial community, and raise the population of antagonistic microorganisms. Plant control measures are to improve the tomato's resistance against bacterial wilt. In summary, the development of a novel environmental protection and efficient control system based on the principles of micro-ecological control and tomato breeding with stronger resistance to bacterial wilt will be the important direction of tomato bacterial wilt control for the time to come.

**Author Contributions:** Conceptualization, H.Y. and Y.L.; validation, S.W., Y.L., F.G., H.S. and H.Y.; investigation, S.W., Y.L., H.Y., X.F., X.Z. and H.S.; resources, Y.L.; writing—original draft preparation, S.W.; writing—review and editing, S.W. and Y.L.; supervision, Y.L. and F.G.; project administration, Y.L. and H.Y.; funding acquisition, Y.L. All authors have read and agreed to the published version of the manuscript.

**Funding:** This work was supported by Ningbo Municipal Science and Technology Bureau (Grant Numbers 2021Z047 and 2021-DST-004).

**Institutional Review Board Statement:** Not applicable.

**Data Availability Statement:** Not applicable.

**Conflicts of Interest:** The authors declare no conflict of interest.

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
