# Peer review of "An Insight into the Prevention and Control Methods for Bacterial Wilt Disease in Tomato Plants"

_agronomy, doi:10.3390/agronomy13123025_

Round 1

Reviewer 1 Report

Comments and Suggestions for Authors

The article is well written in general but needs some corrections, please see on the manuscript.

The manuscript lack tables and figures. I strongly recommend to provide figures showing the mechanisms of different compounds, as shown on the manuscript. I consider that the manuscript would by improved by adding tables and figures. 

There are some confusions about the resistance to phages, or about the genes encoding the resistance, please see on the manuscript.

The section 4.1. Resistant variety should be rewritten. Please organize this section in 2 sections: selection of new resistant varieties and improvement of tomato plants by hybridization and please also include examples about genetic transformation and genome editing. Please include discussion about the advantage of edited plants over genetic transformed ones regarding their acceptance on the market and by consumers. 

Comments on the Quality of English Language

Moderate revision of English language is required.

Author Response

Added images and tables

Resistance to phages has been modified

Section 4.1. Resistant varieties have been adapted.

Reviewer 2 Report

Comments and Suggestions for Authors

The review presents a summarized data on the latest studied and progresses of investigations on tomato bacterial wilt control. The review is organized in a correct sequence and contains comprehensive information on the subject. It includes an assessment of physical and biological methods, application of plant-immune induced resistance, effectiveness of disease management and analysis of the economical aspects. The topic of the manuscript is appropriate for the scope of Agronomy journal.

The paper systematically presents a number of methods for controlling tomato bacterial wilt. Numerous examples from diverse scientific studies are described to support the main idea of the manuscript. The cited literature sources are relevant and most of them have been published recently. Some older publications that contain basic studies on the topic are also used. The successes and weaknesses of various methods, including soil improvement and upgrading plant traits, aimed to overcome the bacterial wilt disease are indicated. The article gives practical recommendations to farmers about the possibilities to minimize the impact of the bacterial wilt or to overcome the disease. All parts of the review and examples support the conclusions that are presented at the end of the manuscript.

Some suggestions and recommendations to the authors are listed below:

·                     The full name of any species should be written when it is mentioned for the first time (Ralstonia solanacearum, etc.). After that the short form should be used (R. solanacearum, ets.). Some names of plats and pathogens are not Italic and should be edited.

·                     English improvement and avoiding repetition of the same word into one sentence or in the next one.

For instance, in one sentence from rows 43-47 phrase “puts forward” is used twice. It is recommended to find out synonyms. All text of the manuscript should be proofread and edited where there are other repetitions.

·                     Row 249 – Is this a correct reference?

·                     Rows 264-266 – The sentence is not clear enough.

Comments on the Quality of English Language

Minor editing of English language required. Recommendations are listed above. 

Author Response

All suggestions have been accepted

Round 2

Reviewer 1 Report

Comments and Suggestions for Authors

The manuscript was improved. All corrections/suggestions were addressed.